# Racial and Economic Disparities in High-Temperature Exposure in Brazil

**DOI:** 10.3390/ijerph22020200

**Published:** 2025-01-30

**Authors:** Hosana Gomes da Silva, Weeberb J. Requia

**Affiliations:** Center for Environment and Public Health Studies, School of Public Policy and Government, Fundação Getulio Vargas, Brasilia 72125590, Brazil; hosanagomes.vet@gmail.com

**Keywords:** weather, extreme temperature, disparities, vulnerable subpopulations

## Abstract

Primary studies analyzing the distribution of exposure to the consequences of climate change among different vulnerable groups are scarce. This study addresses this gap by investigating racial and economic disparities in high-temperature exposure in Brazil, focusing on the impact on vulnerable subpopulations. We utilized georeferenced temperature data from the Global High-Resolution Estimates of Extreme Heat (GEHE) and population data from the 2010 Census. The disparity analyses included (i) estimating the exposure rate to temperatures exceeding 28 °C, expressed as population-weighted heat exposure ($\bar{PHE}$); (ii) determining the difference in exposure between the most and least exposed groups; and (iii) calculating weighted Gini coefficients. The findings reveal that low-income and black, brown, and indigenous populations are predominantly the most exposed to $\bar{PHE}$ exceeding 28 °C. Nationally, the indigenous population is the most exposed racial group, with a $\bar{PHE}$ 47% higher than that of the white population. Stratified analyses indicate that, despite varying climatic and environmental conditions across regions, the black-brown-indigenous population consistently faces the highest heat exposure in Brazil. Income disparity analyses show that the lowest per capita income groups are the most exposed to high temperatures across the country. The study highlights the impact of climate change on economic inequality and the deepening of within-country inequalities, particularly affecting socioeconomically disadvantaged groups. These findings underscore the urgent need for evidence-informed public policies to address racial and economic disparities in high-temperature exposure, mitigate health risks associated with climate change, and emphasize the importance of context-sensitive analyses for a comprehensive understanding of heat-related risks and public health.

## 1. Introduction

The climate regimes across the globe are shifting. The World Economic Forum, in its 2024 Global Risk Report, particularly highlights the process of global warming as one of the most serious current threats to humanity. According to the Intergovernmental Panel on Climate Change (IPCC), the average temperature between 2006 and 2015 was 1.53 °C higher compared to 1850–1900. This increase brings new challenges to the global scenario, including rising sea levels, environmental disasters, and notably, episodes of extreme temperatures known as heatwaves [1].

In the last decade, humanity has witnessed an increase in the frequency and intensity of extreme temperatures. In Brazil, the annual temperature is projected to rise by approximately 1.8 °C by 2100, on average, and it is estimated that the count of high-temperature days will increase from <10 to 90 days if we fail to reduce greenhouse gas emissions [2].

In addition to profound impacts on the terrestrial ecosystem, biodiversity, access to natural resources (such as water), and global food production, extreme temperatures bring to light a growing concern: imminent risks to human health. According to Zhao et al. (2021) [3], approximately 10.34% of global deaths per year can be attributed to inadequate temperatures, resulting in more than 5 million deaths annually. The IPCC indicates that by 2050, climate change will exacerbate pre-existing health problems and may also lead to climate-related diseases in at-risk populations. Research suggests an association between increased mortality and rising temperatures, with such illnesses primarily resulting from cardiovascular diseases and respiratory diseases [4].

Heat exposure has been widely studied in the context of its impacts on health [5], but methods to quantify it vary across the literature. Commonly used metrics include ambient temperature [6,7,8], heat indices [9,10], and composite thermal indices like the Wet Bulb Globe Temperature (WBGT) [11,12,13], which incorporate multiple meteorological variables. Additionally, population-weighted metrics have gained attention for better capturing the exposure burden in densely populated areas [14]. 

These studies have reported that the distribution of health risks due to exposure to extreme temperatures is not equally distributed among all different population groups. According to Paavola (2017) [4], the impacts of climate change on health will not be equal due to differences in exposure, the presence of health determinants, and the adaptive capacity of individuals and groups. The expanded concept of health urges us to view health promotion not only as disease prevention but also to incorporate the defense of fundamental human rights and the reduction of socioeconomic inequities [15]. In this context, climate change raises profound ethical concerns about equity in impacts and responsibilities.

In the literature, studies in the United States, Asia, and Europe have focused on analyzing such disparities [2]. According to Ebi and Hess (2020) [1], “just twenty countries are the source of more than 80% of all current greenhouse gas emissions, with more than 40% of emissions coming from China (28%) and the US (14%) alone”. Manware et al. (2022) [16] developed the Heat Vulnerability Index for the United States and found that communities of color are significantly linked to heightened vulnerability to heat. Specifically, in California, a study found that socioeconomically vulnerable groups are exposed to a greater impact from high temperatures and the concentration of pollutant gases [15]. Consequently, the concept of environmental justice is gaining prominence on the international stage, aiming for equity in addressing climate change to protect vulnerable communities, which, for the most part, are located in regions with the least contribution to global warming.

Diffenbaugh and Burke (2019) [17] state that global warming contributes to worsening global economic inequality. The authors’ estimation suggests that due to the climate crisis, the disparity between high-income and low-income groups has expanded by 25%, compared to what it would have been otherwise. As an example of this reality, a global study conducted in 25 cities across different regions of the world found that in most cities (72%), neighborhoods with lower income levels were the most exposed to heat. Meanwhile, Taconet, Méjean, and Guivarch (2020) [18] identify climate change as one of the main drivers of inequalities, especially as the damage incurred delays economic convergence between poor and rich countries. Studies by Taconet, Méjean, and Guivarch (2020) [18] and Colmer (2021) [19] suggest climate change may also deepen inequalities within countries, as low-income groups face crises in accessing the natural resources necessary for adaptation and survival to climate risks.

The debate on social inequities exposure and health aims to identify populations vulnerable to health risks. In this context, there has been growing attention from the scientific community to discuss the aspects of social justice in the context of climate change adaptations [4]. This is a particularly alarming scenario for Brazil, one of the most unequal countries in the world. According to recent data from the Brazilian Institute of Geography and Statistics [20], the wealthiest 10% of the population earns more than 40 times what the poorest 10% earns, while racial disparities persist across income, education, and employment. Additionally, basic rights such as access to healthcare, education, housing, and nutritious food are disproportionately limited in Brazil. Although studies have explored the vulnerability of socioeconomically disadvantaged populations to heat exposure in other regions [21,22,23,24,25], there is a significant research gap in Brazil, where primary studies analyzing the distribution of exposure to climate change impacts among different vulnerable groups remain scarce. Therefore, this study seeks to address this gap by identifying economic and racial disparities in exposure to high temperatures, identifying groups at risk, and contributing to the formulation of evidence-informed public policies.

## 2. Methods

### 2.1. Weather Data

We used georeferenced temperature data available in the open database Global High-Resolution Estimates of Extreme Heat (GEHE), provided by the NASA Socioeconomic Data and Applications Center (SEDAC). The dataset covers the period from 1983 to 2016 and provides annual global counts on a 0.05-degree grid (~5 km) of the number of days when the Wet Bulb Globe Temperature (WBGTmax) exceeded dangerous thresholds for humid-hot heat for the period from 1983 to 2016. The thresholds are based on the International Organization for Standardization (ISO) criteria for occupational heat-related risk:WBGT > 28 °C: represents moderate heat stress, where physical labor can still occur but with necessary precautions such as hydration and work/rest cycles.WBGT > 30 °C: indicates high heat stress, requiring significant modifications to work practices, such as reduced physical activity and increased rest periods to prevent heat-related illnesses.WBGT > 32 °C: denotes extreme heat stress, where outdoor physical labor is generally unsafe, and even short exposures may result in severe health consequences without protective measures.

GEHE is the highest resolution dataset of its kind to date, and thus, its spatial resolution (approximately 5 km per grid) allowed a thorough analysis of the information in our work. In the context of this study, the analysis was focused on the year 2016, which represents the most updated data available in the database. We considered days when the temperature exceeded 28 °C, a threshold already recognized as indicative of risk to human health, as mentioned earlier.

### 2.2. Population Data

We used population data (race and income) available in the 2010 Census, provided by the Brazilian Institute of Geography and Statistics (BIGS). These data are spatially resolved at the Census tract level. The “census tract” is a geographic unit used to collect and analyze data in population censuses and other demographic surveys. It is a continuous area, located within a single urban or rural area, with dimensions and number of households that allow for surveying by a census enumerator. This is the finest scale of available population data in Brazil. We accessed data from 316,514 census tracts covering the Brazilian territorial area of 8,515,692.27 km^2^.

We analyzed population data based on race and income. According to BIGS, racial data are constructed through the self-declaration of each individual as belonging to one of the following groups: black, pardo (mainly used to refer to people of light brown skin color), white, indigenous, or yellow (BIGS’s designation for individuals of Asian origin). We used the absolute number of individuals who self-declared belonging to each group within each census tract. 

Regarding income data, we used the average monthly per capita income (value in Brazilian real) for each census tract. The per capita income is calculated as the sum of the total monthly income value of all families in the census tract divided by the total number of individuals in these families. To calculate, we identified census tracts with low income as those in the bottom quartile, i.e., the 25% with the lowest average per capita income values. Similarly, we defined census tracts with high income as those situated in the upper quartile, representing the 25% with the highest values.

### 2.3. Spatial Aggregation

Initially, we spatially aggregated temperature data with race and income population data at the census tract level. For this, we considered the average WBGTmax value of the geographic limits (polygon) of each census tract. The resulting database is composed of the following information: the number of days in 2016 in which that specific census tract experienced humid-heat levels above 28 °C, the number of self-declared individuals in each racial group separately, and the average monthly per capita income value of families. We used this resulting database for the disparities analyses.

### 2.4. Disparities Analyses

The disparity analyses were performed in three steps: (i) estimating the exposure rate to temperature exceeding 28 °C expressed as population-weighted heat exposure ($\bar{PHE}$), (ii) estimation of the difference between exposure for the most exposed group versus the least exposed group, and (iii) estimation of the weighted Gini coefficients. All of these steps were initially conducted for the distribution of exposure to high temperatures at the national level, and subsequently, the same analyses were performed by stratifying the data according to the five Brazilian regions (North, Northeast, Midwest, Southeast, and South—Figure 1 shows the spatial distribution of these Brazilian regions).

#### 2.4.1. Population-Weighted Heat Exposure

The population-weighted heat exposure was calculated within two groups—racial and income-based. The heat exposure of racial group k at the national level was computed as depicted in the following equation:(1)$${\bar{PHE}}_{k}=\frac{\sum_{j=1}^{n}{PHE}_{j}P_{k,j}}{\sum_{j=1}^{n}P_{k,j}}$$
where ${\bar{PHE}}_{k}\;$ represents the population-weighted heat exposure (PHE) at a national level, of racial group *k* (white, black, pardo, Asian, or indigenous); PHE*_j_* is the number of days the racial group *k* experienced temperatures exceeding 28 °C in census tract *j*; *P_k_*_,*j*_ is the number of self-declared individuals belonging to racial group *k* residing in census tract *j*; and *n* represents the total number of census tracts in Brazil. For the income groups, the population-weighted heat exposure was calculated as follows:(2)$${\bar{PHE}}_{i}=\frac{\sum_{j=1\;(j\;\in \;i)}^{n}{PHE}_{j}P_{j}}{\sum_{j=1\;(j\;\in \;i)}^{n}P_{j}}$$
where $\;{\bar{PHE}}_{i}\;$ is the national population-weighted heat exposure for income group *i*, expressed as the number of days the income group *i* experienced temperatures exceeding 28 °C. We divided the dataset into two income groups, including a group characterized as low income (<25th quartile) and a group defined as high income (>75th quartile). PHE*_j_* is the number of days when ambient temperatures exceeded 28 °C in census tract *j*; and *P_k_*_,*j*_ represents the total inhabitants in census tract *j*. Note that the summation is conducted only across census tracts *j* within income group *i*. Therefore, *n* denotes the total number of these census tracts.

#### 2.4.2. Difference in Exposure Between the Most Exposed Group and the Least Exposed Group

We assessed the discrepancy in exposure based on three distinct measures: absolute disparity, percent difference, and relative disparity (or ratio). The absolute difference was calculated by the difference between the population-weighted heat exposure of the most exposed group (racial and income) and that of the least exposed group (racial and income). For example, if the Black population exhibits higher exposure to extreme heat days than the White population (${\bar{PHE}}_{black}$ > ${\bar{PHE}}_{white}$), the absolute disparity will be calculated as ${\bar{PHE}}_{black}$ − ${\bar{PHE}}_{white}$. As for the second measure (percent difference), still considering the same example (${\bar{PHE}}_{black}$ > ${\bar{PHE}}_{white}$), the percent difference is calculated as [(${\bar{PHE}}_{black}$ − ${\bar{PHE}}_{white}$)/national average of the number of days in which ambient temperatures exceeded 28 °C] × 100%. Finally, the relative disparity is determined by the ratio ${\bar{PHE}}_{black}/{\bar{PHE}}_{white}$. Note that the percent difference and relative disparity measures are used to quantify disproportionality in exposure burdens [26]. Finally, we conducted a sensitivity analysis in which we excluded data from the indigenous group, as it resides predominantly in specific geographic regions with unique environmental conditions, such as the Amazon rainforest. These conditions could disproportionately influence the overall results. 

#### 2.4.3. Estimation of the Weighted Gini Coefficients and Sensitivity Analysis

The previously described steps fail to capture disparities across the entire distribution of days in which temperature exceeded 28 °C, given that the approach was based on the population-weighted mean number of days in which temperature exceeded 28 °C. To address this limitation and validate the consistency of our initial findings (as discussed in the previous section), an additional measure of inequality was calculated in this subsequent stage. This measure considers the complete distribution of exposure to extreme heat days, estimating weighted Gini coefficients for each racial group and the overall population. To perform these calculations, we employed the “weighted.gini” function from the R package “acid”. This function was provided with the exposure variable and the population of each racial group (as well as the total population) in each census tract.

## 3. Results

Table 1 shows the geographic composition of Brazilian racial groups across the country. Nationally, the Brazilian population consists of 43.5% white individuals, 10.1% black individuals, 45.3% pardo individuals, 0.8% indigenous individuals, and 0.4% individuals of Asian origin. Historically, the racial distribution in Brazil undergoes constant changes mainly due to the characteristic of self-declaration of race. Therefore, currently, the proportion of Brazilians identifying themselves as belonging to the historically marginalized black-pardo-indigenous group represents 56.1%. Observing the geographic distribution of these groups, Figure 1 indicates that the largest concentration of black-brown-indigenous individuals is concentrated in the North and Northeast regions. The highest-income population (Table 2) and the largest proportion of the white and Asian population in the country are concentrated in the Midwest, Southeast, and South. 

Table 3 provides a descriptive summary of the variable temperature, represented by the number of days with temperatures exceeding 28 °C, across different Brazilian regions. Nationally, the mean number of such high-temperature days is 39.49, with a wide standard deviation of 73.01, indicating significant variability across the country. The minimum and maximum values range from 0.042 to 349 days. Regionally, the North experiences the highest mean number of hot days at 205.95, with a relatively smaller standard deviation of 57.92. This region also has the highest median value (215 days) and interquartile range, suggesting consistent high-temperature exposure. The Northeast has a mean of 53.42 days, but with a high standard deviation of 88.62, reflecting considerable variability. The Midwest region reports a mean of 40.80 days, with a standard deviation of 55.47, while the Southeast and South regions exhibit much lower mean values of 12.85 and 7.78 days, respectively, and smaller standard deviations. These statistics highlight the substantial regional disparities in high-temperature exposure, with the North region being particularly affected, followed by the Northeast and Midwest, while the Southeast and South regions experience comparatively fewer high-temperature days.

At the national level (Table 4), the indigenous population is the most exposed group to high temperatures, while the least exposed group is the white population (Table 4). Among the racial groups studied, the indigenous population has the highest level of heat exposure, with an average exposure ($\bar{PHE}$) of 87.19, which is significantly higher (147.96%) than that of the white population ($\bar{PHE}$ = 28.76). Regionally, the North region shows that the pardo population is the most exposed to high temperatures ($\bar{PHE}$ = 215.17), followed by the indigenous population ($\bar{PHE}$ = 175.35), with an absolute disparity of 39.82 and a relative disparity of 1.22. In the Northeast, the pardo population again leads in exposure ($\bar{PHE}$ = 63.11), followed by the indigenous population ($\bar{PHE}$ = 48.87), with an absolute disparity of 14.24 and a relative disparity of 1.29. In the Midwest, the indigenous population is most exposed ($\bar{PHE}$ = 56.07), while the Asian population is least exposed ($\bar{PHE}$ = 30.83), resulting in an absolute disparity of 25.23 and a relative disparity of 1.81. The Southeast region shows the black population as the most exposed ($\bar{PHE}$ = 75.23), with the Asian population being the least exposed ($\bar{PHE}$ = 10.83), yielding the highest absolute disparity of 64.39 and a substantial relative disparity of 6.94. In the South, the black population is also the most exposed ($\bar{PHE}$ = 9.38) compared to the indigenous population ($\bar{PHE}$ = 5.96), with an absolute disparity of 3.41 and a relative disparity of 1.57 (Table 4).

Regarding income disparities (Table 4), at the national level, the population in the first quartile of the income distribution (Q25) is the most exposed to high temperatures ($\bar{PHE}$ = 78.23), while those in the third quartile (Q75) are the least exposed ($\bar{PHE}$ = 27.23). This results in an absolute disparity of 50.99 and a relative disparity of 2.87. This result, where the lowest-income population group was the most exposed, persisted in the subgroup analyses by region.

For a better comparison of the exposure among the racial and income groups, we illustrate the $\bar{PHE}$ for all groups in a chart shown in Figure 2, which also allows us to see the ranking of exposure among the groups.

The Gini coefficients provide an overview of the inequalities present in each racial/income group. The Gini coefficient remained above 0.50 in all analyses. The North region had the lowest Gini, varying from 0.13 to 0.16 among the racial groups. At the national level, the white population exhibited the highest Gini coefficient (0.79), while for the black, Asian-origin, pardo, and indigenous populations, the coefficients were 0.74, 0.75, 0.70, and 0.57, respectively (Table 5).

We conducted a sensitivity analysis in which we excluded data from the indigenous group, the group with the smallest population, to determine whether this exclusion would significantly impact the outcomes. The indigenous population in Brazil is relatively low compared to other groups. Following the exclusion of this demographic subgroup, the pardo population emerged as the most exposed group at the national level. Upon regional analysis, the black-pardo-indigenous population consistently exhibited the highest levels of exposure across most regions. Notably, the results for the Northeast region diverged from those of the primary analysis, with the Asian population demonstrating the highest level of population-weighted heat exposure.

## 4. Discussion

Our study found that black-pardo-indigenous and low-income individuals faced the greatest impact from heightened population-weighted heat exposure across Brazil. This pattern of inequality persists consistently across the Brazilian regions. Overall, the white population at the national level has the lowest exposure (three times lower when compared to the indigenous population, which has the highest rate; two times lower compared to pardos, with the second highest rate; and 1.66 and 1.61 times lower when compared to Asian-origin and black populations, respectively). 

Upon examining the groups identified as the least exposed, the results varied in the regional analysis. Overall, at the national level, the white population has the lowest exposure, in the North, Northeast, and South regions, while the indigenous population was found to be the least exposed. In the Midwest and Southeast regions, the population of Asian origin exhibited the lowest heat exposure. We highlight that in areas characterized by dense urbanization, such as the Southeast, Northeast, and Midwest regions, the white population emerged as the second least exposed demographic group.

The discrepancy observed in the results between the primary analysis and the sensitivity analysis, in which the Asian population emerged with the highest population-weighted heat exposure, points to the importance of carefully examining regional and demographic nuances when assessing the impacts of environmental conditions. Possible contributing factors to this discrepancy may include differences in the geographical distribution of the Asian population, patterns of urbanization, and specific regional climate characteristics. These findings underscore the complexity of the interaction between socioeconomic, demographic, and environmental factors in determining heat exposure patterns, highlighting the ongoing need for context-sensitive analyses for a comprehensive understanding of heat-related risks and public health.

Our findings are in agreement with studies conducted in Europe [27,28], Asia [29,30], the United States [16,26,31,32], and globally [17,18,33]. All these previous studies suggest that vulnerable populations that live with limited access to quality basic resources, and consequently, having lower adaptive capacity to climate change, are also the most exposed to the effects of this process. This is particularly true in the United States, where [32] examined racial disparities in mortality across four cities. They found that the black population had a 5.3% higher heat-related mortality compared to the white population within the same temporal and spatial contexts. Another study in the United States developed the Heat Vulnerability Index (HVI) and found that in the census tracts with the highest HVI scores, 75.6% of the population was non-white, while in those with the lowest scores, 24.7% were non-white [16]. A study across 25 cities globally (including the Brazilian city São Paulo) reported that in the majority of cities (72%), neighborhoods with lower income levels were predominantly the most exposed to heat. The study highlights that the primary factor of intra-urban variability in this exposure is the neighborhood’s vegetation density, among other environmental characteristics [33]. Following this context, the European Environment Agency (EEA) investigated the social distribution of environmental risks. It identified socioeconomically disadvantaged groups as the most susceptible to high temperatures, alongside other risk factors such as noise and air pollution [27]. In the Netherlands, an analysis of land surface temperature found an association between higher property values and reduced exposure to high temperatures, in which residents with higher incomes, acquiring pricier properties in cooler areas, experienced significantly less heat exposure [28].

Based on the previous literature, we suggest that in urban areas, one of the primary factors driving intra-urban variability in heat exposure is the neighborhood’s vegetation density. Chakraborty et al. (2019) [33] demonstrated that neighborhoods with lower income levels typically have less vegetation cover, contributing to greater heat exposure through the urban heat island effect. This aligns with evidence from our study showing higher exposure rates in densely populated urban regions like the Southeast and Northeast, where vulnerable groups often reside in poorly vegetated neighborhoods. In less urban areas, where the urban heat island effect is less pronounced, heat exposure differences may be influenced by other environmental factors. For example, proximity to water bodies can create cooler microclimates due to evaporative cooling, while areas with higher elevations tend to have lower temperatures. Additionally, regional climatic differences, such as humidity levels and prevailing wind patterns, may also explain the variation. Future studies should incorporate these factors to better understand the drivers of heat exposure in non-urban settings. 

Brazilian studies investigating environmental injustice are still scarce. Some research has assessed disparities in certain cities or specific regions of the country. In the state of São Paulo, located in the Southeast region, Pereira, Masiero, and Bourscheidt (2021) [34] examined socio-spatial inequality and its relationship to thermal discomfort. They found that buildings located along the seafront in the city of Santos generally provide residents with comfortable air temperatures, benefiting from shading and sea breezes. These residences are typically occupied by high-income individuals. In contrast, higher average air temperatures were identified in the city’s peripheral regions, where a higher concentration of low-income residents live in precarious urban environments. Another Brazilian study found different results compared to our findings. Requia and Castelhano (2023) [35] estimated the economic and racial disparities of the weather impact on air quality in Brazil between 2003 and 2018. They reported that high-income and white populations were the most exposed groups. The authors suggest that these findings may be explained by Brazil’s complex environmental and socio-demographic conditions. 

Chisadza et al. (2023) [36] underscore three pathways through which climate change can deepen inequities for vulnerable groups. First, they highlight that local exposure to climate change effects may exacerbate existing inequalities within already unequal countries. For instance, Chisadza et al. (2023) [36] cite New Orleans, where an increasing concentration of low-income African American residents was observed in districts severely affected by Hurricane Katrina. Second, they note that increased vulnerability arises from a lack of access to resources. Third, there is a reduced capacity among vulnerable groups to cope with and recover from climate change-induced damages, which exacerbates their initial income inequality. 

In this line, Jacobi and Sulaiman (2016) [37] highlight a strong social dimension within the construction of adaptive capacity, emphasizing that adaptation measures depend on the ability of populations to identify risks and promptly respond to the effects of climate change, including the prevention and minimization of consequences. According to the authors, in Brazil, inequality directly affects the social distribution of risks due to the lack of access to basic rights such as housing, healthcare, education, and nutrition, which results in greater vulnerability to the effects of the climate crisis.

Economic studies on racial disparity in Brazil reveal that countries which extensively relied on enslaved labor from human trafficking (mainly from Africa during their colonization) now face significant socioeconomic vulnerabilities for their black and pardo populations [38]. Data from the Brazilian Institute of Geography and Statistics [20] supports this finding, highlighting that Brazil, the last country in the Americas to formally abolish slavery, exemplifies this issue. In Brazil, 70% of management positions are held by white individuals, while black and pardo people make up approximately 75% of the population within the poorest 10% of the country. This discriminatory scenario is also supported by income and employability data, showing that black and pardo workers in Brazil earn, on average, 14.25% less than their white counterparts with similar education, experience, and employment status [38].

The Asian population in Brazil, though small (0.4% of the national population), has a unique socioeconomic and historical context. Many Asians in Brazil, particularly those of Japanese descent, arrived as immigrants in the early 20th century and are often concentrated in urban areas. The Asian population tends to have a relatively high socioeconomic status (SES), with access to higher education and stable employment opportunities. However, these generalizations do not uniformly apply to all Asian subgroups, as recent migrants from countries like China and Korea may face different economic challenges, including integration into the labor market and language barriers. This heterogeneity in the Asian population’s SES highlights the importance of nuanced analyses when addressing racial disparities in Brazil. 

In this context, the Oswaldo Cruz Foundation (Fiocruz), a Brazilian institution dedicated to Science and Technology in Health, analyzed empirical evidence regarding disparities in exposure to environmental conflicts and termed these disparities as environmental injustices. The pioneering study “Environmental Injustice and Health in Brazil”, conducted in 2013, demonstrates that these injustices predominantly impact the health of poor, indigenous, black, and pardo populations in Brazil. Endorsed by United Nations Development Programme (UNDP) technical papers, environmental injustice relates to the concern that environmental risks and hazards disproportionately affect societal groups in the most vulnerable and less empowered contexts and countries [39]. Additionally, studies underscore that denying the existence of discrimination leads to this phenomenon being interpreted solely as a consequence of economic and class disparities, thereby harming social development by neglecting to address the daily discrimination faced by non-white individuals [40].

This study presents several strengths that contribute significantly to understanding environmental injustice in Brazil. First, to our knowledge, it stands out as the first comprehensive examination of economic and racial disparities in exposure to high temperatures across the entire Brazilian territory. This broad scope allows for a more nuanced and complete understanding of the challenges faced by different population groups. Second, the use of data obtained at the census tract level provides a granular view of the population distribution, enabling a detailed analysis of disparities at a localized level. This approach adds depth to the study by capturing variations within regions and cities, which may not be evident in broader analyses. Finally, the study employs multiple metrics of disparities, enriching the analysis and providing a comprehensive picture of the complexities involved in environmental injustice in Brazil.

However, it is important to acknowledge certain limitations. First, despite the detailed analysis at the census tract level, there may still be limitations in the representativeness of these areas for the entire population. Second, while using multiple metrics enhances the understanding of disparities, it may also introduce complexities in interpretation and comparison with other studies. Third, the study’s reliance on historical data may limit the generalizability of the findings to current or future scenarios, considering potential changes in population demographics or urban development patterns. Another limiting aspect is that the population data used correspond to the 2010 Population Census, the most recent data available at the time of this research. While the weather data were derived from 2016, using population data from the 2010 Census introduces a temporal gap in the study, potentially limiting the accuracy and relevance of the findings to the current population landscape. We assumed that the spatial distribution of population characteristics did not change significantly between 2010 and 2016. This assumption is supported by historical trends showing gradual demographic changes in Brazil. Additionally, our focus on spatial patterns rather than precise temporal trends helps mitigate concerns related to the mismatch in years. Finally, the racial information of individuals was derived from self-reporting. While self-declaration is a common method for gathering racial data, race is a sociological concept; therefore, self-declaration introduces a potential source of bias because it is subject to personal interpretation. This can be particularly variable in contexts where racial categories are fluid. It is important to highlight the subjective nature of self-reporting and underscore this aspect when interpreting racial data collected.

## 5. Conclusions

Our study suggests significant disparities, with marginalized communities, including racial minorities and lower-income groups, experiencing higher levels of heat exposure compared to wealthier and non-minority populations. Nationally, the indigenous population was identified as the most exposed group, with a population-weighted heat exposure ($\bar{PHE}$) that was 47% higher than that of the white population. Similarly, individuals in the lowest income quartile faced nearly three times the exposure compared to those in the highest income quartile, underscoring how socioeconomic and racial vulnerabilities intersect to amplify climate-related risks. These findings underscore the intersectional nature of environmental justice issues, where vulnerable communities bear a disproportionate burden of climate-related risks. The findings of this study have significant implications for environmental and public policy in Brazil and beyond.

From an environmental perspective, our study underscores the need for targeted interventions to mitigate heat-related risks in vulnerable communities. This may include urban planning strategies such as increasing green spaces, implementing heat-resilient infrastructure, and promoting climate-adaptive housing designs in low-income neighborhoods. Our findings suggest that regional disparities in heat exposure should also guide climate adaptation policies, particularly in the North and Northeast regions, where heat exposure levels are significantly higher. Investments in reforestation, improved access to water resources, and enhanced local climate monitoring could help address region-specific vulnerabilities. Incorporating heat-mitigation measures into broader climate action plans can help reduce the health impacts of extreme heat events and enhance the overall resilience of communities.

On the public policy front, our findings highlight the importance of integrating equity considerations into policymaking processes. Policies aimed at reducing heat exposure disparities should prioritize the needs of marginalized groups, ensuring access to affordable cooling options, healthcare services, and social support during heatwaves. Given the rapidly changing climate, there is a need for dynamic and regularly updated heat vulnerability assessments to account for evolving demographic and environmental conditions. Additionally, our results suggest that a deeper focus on indigenous communities is crucial to addressing the disproportionate risks they face, especially considering their unique geographic and cultural contexts. Implementing inclusive policies that address social determinants of health, such as income inequality and access to quality housing, can contribute to building more resilient and equitable communities in the face of climate change.

## Figures and Tables

**Figure 1 ijerph-22-00200-f001:**
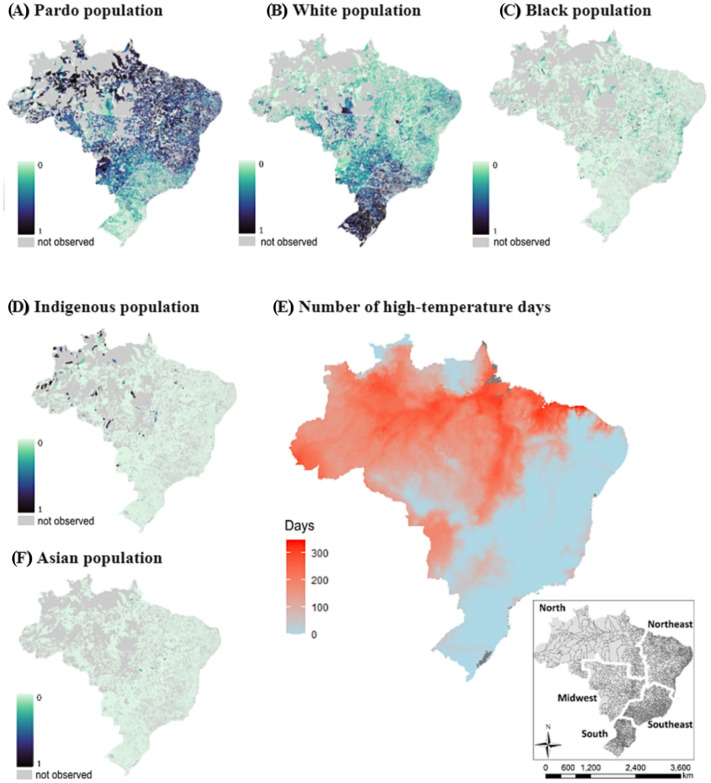
Distribution of race and heat across Brazil. Note 1: 0–1 represents the proportion of each racial group. Note 2: “Not observed” indicates locations with no self-declared affiliation to that race.

**Figure 2 ijerph-22-00200-f002:**
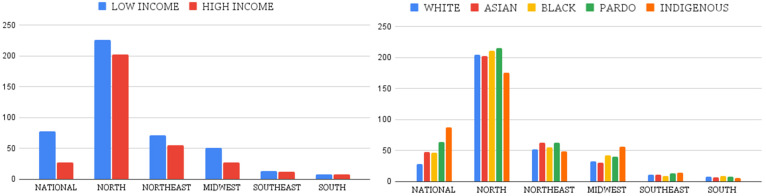
Population-weighted heat exposure ($\bar{PHE})$ by race and income.

**Table 1 ijerph-22-00200-t001:** Percentage (%) of each racial group by region.

	White	Black	Asian	Pardo (or Brown)	Indigenous
National (Brazil)	43.50	10.10	0.40	45.30	0.80
North	20.70	8.80	0.20	67.20	3.10
Northeast	26.70	13.00	0.10	59.60	0.60
Midwest	37.00	9.20	0.40	52.40	1.00
Southeast	49.90	10.60	0.70	38.70	0.10
South	72.60	5.00	0.40	21.70	0.30

**Table 2 ijerph-22-00200-t002:** Descriptive statistics the variable income (BRL) across Brazilian regions.

	Min	Q1	Median	Mean	SD	Q3	Max
National (Brazil)	0.11	348.92	542.515	750.72	1638.44	825.84	50,445.06
North	0.59	249.12	386.21	515.26	685.07	584.86	48,093.81
Northeast	1.22	208.66	292.53	434.57	514.16	450.63	12,659.84
Midwest	0.39	479.39	631.36	886.03	890.57	920.64	26,765.26
Southeast	0.11	468.71	639.56	913.07	906.15	965.16	30,825.93
South	0.62	510.00	688.13	900.28	3,471.14	975.24	504,450.56

Note: Standard deviation (SD), minimum (Min), maximum (Max), first quartile (Q1), and third quartile (Q3).

**Table 3 ijerph-22-00200-t003:** Descriptive statistics for the variable temperature used in the analyses across Brazilian regions. Temperature represents the number of days with temperature exceeding 28 °C.

	Min	Q1	Median	Mean	SD	Q3	Max
National (Brazil)	0.04	5.00	19.00	39.49	73.01	57.00	349.00
North	0.25	173.00	215.00	205.95	57.92	251.00	319.00
Northeast	0.04	3.50	13.00	53.42	88.62	76.00	349.00
Midwest	0.04	12.00	55.00	40.80	55.47	117.17	237.20
Southeast	0.06	4.00	24.00	12.85	18.76	39.00	97.00
South	0.71	4.00	10.00	7.78	8.98	15.00	56.00

Note: Standard deviation (SD), minimum (Min), maximum (Max), first quartile (Q1), and third quartile (Q3).

**Table 4 ijerph-22-00200-t004:** Difference between the population-weighted heat exposure for the most exposed group (racial and income group) versus the least exposed group. Results from the main analysis.

	Region	Most Exposed Group	Least Exposed Group	Absolute Disparity	% Difference	Relative Disparity
Racial	National	Indigenous ($\bar{PHE}$ = 87.19)	White ($\bar{PHE}$ = 28.76)	58.43	147.96	3.03
	North	Pardo ($\bar{PHE}$ = 215.17)	Indigenous ($\bar{PHE}$ = 175.35)	39.82	19.33	1.22
	Northeast	Parda ($\bar{PHE}$ = 63.11)	Indigenous ($\bar{PHE}$ = 48.87)	14.24	26.66	1.29
	Midwest	Indigenous ($\bar{PHE}$ = 56.07)	Asian ($\bar{PHE}$ = 30.83)	25.23	61.84	1.81
	Southeast	Black ($\bar{PHE}$ = 75.23)	Asian ($\bar{PHE}$ = 10.83)	64.39	501.16	6.94
	South	Black ($\bar{PHE}$ = 9.38)	Indigenous ($\bar{PHE}$ = 5.96)	3.41	43.91	1.57
Income	National	Q25 ($\bar{PHE}$ = 78.23)	Q75 ($\bar{PHE}$ = 27.23)	50.99	129.14	2.87
	North	Q25 ($\bar{PHE}$ = 226.45)	Q75 ($\bar{PHE}$ = 202.98)	23.47	11.39	1.11
	Northeast	Q25 ($\bar{PHE}$ = 71.90)	Q75 ($\bar{PHE}$ = 55.77)	16.13	30.20	1.28
	Midwest	Q25 ($\bar{PHE}$ = 51.04)	Q75 ($\bar{PHE}$ = 27.35)	23.68	58.04	1.86
	Southeast	Q25 ($\bar{PHE}$ = 13.66)	Q75 ($\bar{PHE}$ = 12.49)	1.16	9.06	1.09
	South	Q25 ($\bar{PHE}$ = 7.90)	Q75 ($\bar{PHE}$ = 7.76)	0.14	1.84	1.01

Note: population-weighted heat exposure ($\bar{PHE}$), third quartile of the income distribution (Q75), and first quartile of the income distribution (Q25).

**Table 5 ijerph-22-00200-t005:** Gini coefficients for the total population and racial group.

Spatial Level	Total Population	White Group	Black Group	Yellow (or Asian) Group	Pardo (or Brown) Group	Indigenous Group
National	0.76	0.80	0.75	0.75	0.71	0.57
Midwest	0.72	0.73	0.71	0.77	0.71	0.52
Northeast	0.72	0.74	0.74	0.70	0.72	0.72
North	0.14	0.16	0.15	0.15	0.13	0.20
Southeast	0.72	0.74	0.66	0.77	0.69	0.68
South	0.59	0.58	0.49	0.64	0.65	0.68

## Data Availability

The original contributions presented in the study are included in the article; further inquiries can be directed to the corresponding author.

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
