# Peer review of "Racial and Economic Disparities in High-Temperature Exposure in Brazil"

_ijerph, 2025, doi:10.3390/ijerph22020200_

Round 1

Reviewer 1 Report

Comments and Suggestions for Authors

Thank you very much for the opportunity to review the manuscript, “Racial and economic disparities in high-temperature exposure in Brazil.” As extreme heat becomes a more pressing public health issue due to climate change, it is important to identify those communities that are more vulnerable to exposure in order to prevent adverse consequences and address racial/ethnic and socioeconomic health disparities. The manuscript is very well written, and the authors do a great job clearly explaining very complex methodology. I believe the following revisions will make this already great manuscript suitable for publication in IJERPH:

1.        The authors write that Brazil is one of the most unequal countries in the world. They should not only cite this statement, but also perhaps give readers a broad strokes overview of the country’s socioeconomic composition for those not as familiar with the local context. 

2.        The authors write that occupational heat-related risk corresponds to a max WBGT greater than 28, 30, and 32 degrees Celsius. What degree of danger does each of these temperatures correspond to? 

3.        The authors write that they considered the average max WBGT of the “headquarters” of each census tract. I’m not entirely clear what they mean by the term “headquarters” here.

4.        The authors have a typo on Line 177. I believe the term at the beginning of the line should be PHEwhite.

5.        The authors conducted a sensitivity analysis removing the indigenous population because it had the smallest population. Based on Table 1, however, it appears as though Asian was the smallest racial/ethnic group.

6.        The authors write in the discussion that the primary factor of intra-urban variability in heat exposure is neighborhood vegetation. I would recommend that they include the data supporting this statement somewhere in the Results or a Supplement because it’s an important explanatory mechanism. What do they think could explain heat exposure differences in less urban areas? 

7.        The authors do a nice job of explaining how the legacies of slavery and colonialism render black/pardo/indigenous communities more vulnerable to heat exposure. It would be helpful if they could also situate Asians within the socioeconomic and political Brazilian context for those readers less familiar with the country’s population. Do Asians tend to be of higher or lower SES, and is this uniform among all Asian groups? 

Reviewer 2 Report

Comments and Suggestions for Authors

Here are my comments about the manuscript titled ‘Racial and economic disparities in high-temperature exposure in Brazil’.

(1)   Part Introduction - some important references in this part are missing, please check it and cite them correctly. In addition, check the reference style. Moreover, the literature review of heat exposure needs to be included, especially how to quantify it. Furthermore, to the best of my knowledge, there are several works have already explored vulnerable groups, please emphasized the research gap.

(2)   It is good to add a figure to introduce the study area, because the subsequent analysis mentions different regions, e.g., North, and Northeast.

(3)   Line 108-110, can you provide more details that 28 °C is reasonable?

(4)   The data used in this study were derived from different years, how was the fusion analysis of different years carried out? How can the accuracy of the results be assured if the years do not match?

(5)   Part Results - why the days for 0.042 (Line 215)? It seems unreasonable. In addition, most of the analyses are regular, please improve it.

(6)   More fresh findings can be introduced in the Part Conclusion.

Round 2

Reviewer 2 Report

Comments and Suggestions for Authors

I have no further questions.